# ENAT: Rethinking Spatial-temporal Interactions in Token-based Image Synthesis

**Zanlin Ni**[1*]    **Yulin Wang**[1*]    **Renping Zhou**[1]    **Yizeng Han**[1]
**Jiayi Guo**[1]    **Zhiyuan Liu**[1]    **Yuan Yao**[2†]    **Gao Huang**[1†]
[1]Tsinghua University    [2]National University of Singapore

## Abstract

Recently, token-based generation approaches have demonstrated their effectiveness in synthesizing visual content. As a representative example, non-autoregressive Transformers (NATs) can generate decent-quality images in just a few steps. NATs perform generation in a progressive manner, where the latent tokens of a resulting image are incrementally revealed step-by-step. At each step, the unrevealed image regions are padded with `[MASK]` tokens and inferred by NAT, with the most reliable predictions preserved as newly revealed, visible tokens. In this paper, we delve into understanding the mechanisms behind the effectiveness of NATs and uncover two important interaction patterns that naturally emerge from NAT's paradigm: *Spatially* (within a step), although `[MASK]` and visible tokens are processed uniformly by NATs, the interactions between them are highly asymmetric. In specific, `[MASK]` tokens mainly gather information for decoding. On the contrary, visible tokens tend to primarily provide information, and their deep representations can be built only upon themselves. *Temporally* (across steps), the interactions between adjacent generation steps mostly concentrate on updating the representations of a few critical tokens, while the computation for the majority of tokens is generally repetitive. Driven by these findings, we propose EfficientNAT (ENAT), a NAT model that explicitly encourages these critical interactions inherent in NATs. At the spatial level, we *disentangle* the computations of visible and `[MASK]` tokens by encoding visible tokens independently, while decoding `[MASK]` tokens conditioned on the fully encoded visible tokens. At the temporal level, we prioritize the computation of the critical tokens at each step, while maximally *reusing* previously computed token representations to supplement necessary information. ENAT improves the performance of NATs notably with significantly reduced computational cost. Experiments on ImageNet-256$^2$ & 512$^2$ and MS-COCO validate the effectiveness of ENAT. Code and pre-trained models will be released at `https://github.com/LeapLabTHU/ENAT`.

## 1 Introduction

Recent years have witnessed an unprecedented growth in the field of AI-generated content (AIGC). In computer vision, diffusion models [10, 59, 61] have emerged as an effective approach. On the contrary, within the context of natural language processing, content is typically synthesized via the generation of discrete tokens using Transformers [72, 19, 5, 55]. Such discrepancy has excited a growing interest in exploring token-based generation paradigms for visual synthesis [7, 85, 33, 87, 6, 35]. Different from diffusion models, these approaches utilize a discrete data format akin to language models. This makes them straightforward to harness well-established language model optimizations such as the

---

*Equal contribution.
†Corresponding authors.

refined scaling strategies [5, 54, 31, 73] and the progress in model infrastructure [65, 12, 8, 34, 96]. Moreover, explorations in this field may facilitate the development of more advanced, scalable multimodal models with a unified token space [17, 68, 18, 44, 90] as well as general-purpose vision foundation models that integrate visual understanding and generation capabilities [35, 69].

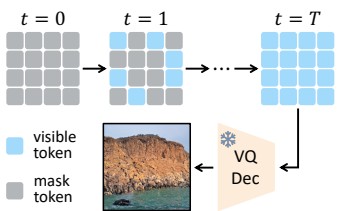

The recent advances in token-based visual generation have seen the rise of non-autoregressive Transformers (NATs) [7, 33, 6, 53], which are distinguished by their abilities to fulfill efficient and high-quality visual synthesis. As shown[3] in Figure 1, NATs follow a progressive generation paradigm: at each generation step, a certain number of latent tokens of the resulting image are decoded in parallel, and the model carries out this process iteratively to produce the final complete token maps. More specifically, at each step, the unknown latent tokens of the image are represented with [MASK] tokens and concatenated with the tokens that have been decoded (*i.e.*, visible tokens). Then, the full set of [MASK] and visible tokens is fed into a Transformer-based model, predicting the proper values of the unknown tokens, with the most reliable predictions preserved as the increments of visible tokens for the next generation step.

Figure 1: **The generation process of NATs** starts from a masked canvas, decode multiple tokens per step, and are then mapped to the pixel space using a pre-trained VQ-decoder [13].

In this paper, we seek to advance the understanding of the mechanisms behind the effectiveness of NATs' progressive generation procedures. Our investigation uncovers two important findings regarding the *spatial* and *temporal* interactions within NATs: *Spatially*, at each generation step, even though both [MASK] and visible tokens are treated equivalently within the computational graphs of NATs, the visible tokens naturally learn to mainly provide information for [MASK] tokens to infer the unknown image content, and their corresponding deep representations can be built in the absence of [MASK] tokens. *Temporally*, the interactions between adjacent generation steps mainly concentrate on updating the representations of a small number of "critical tokens" on top of the previous steps. In fact, the computation for the remaining majority of tokens is generally repetitive.

Inspired by these findings, we propose to develop novel NAT models to explicitly encourage these critical interaction mechanisms emerged naturally when trained for visual generation, yielding EfficientNAT (ENAT). Specifically, at the *spatial* level, we *disentangle* the computations of visible and [MASK] tokens by encoding visible tokens independently of [MASK] tokens. [MASK] tokens are then processed by attending to the fully contextualized features of visible tokens, as shown in Figure 3b. As an interesting observation derived from disentanglement, we find that prioritizing the computation for visible tokens, particularly when the computation is maximized for visible tokens and minimized for [MASK] tokens (even with only a single network layer), further improves the performance of NATs by a large margin. At the *temporal* level, we concentrate computation on the "critical tokens" while maximally *reusing* the representation of previously computed tokens to supplement the necessary information, as illustrated in Figure 4b.

Empirically, the effectiveness of ENAT is validated on ImageNet 256×256 [60], ImageNet 512×512 [60] and MS-COCO [36]. ENAT is able to achieve significantly reduced computational cost compared to conventional NATs while outperforming them notably (*e.g.*, 24% relative improvement with 1.8× lower cost, see Table 6a).

## 2   Related Work

**Image tokenizer and token-based image generation models.** Language models use algorithms like Byte Pair Encoding or WordPiece to convert text into tokens. Similarly, an image tokenizer transforms images into visual tokens for token-based image generation. Key works in this field include Discrete VAE [58], VQVAE [71], and VQGAN [13], with VQGAN-based tokenizers being most popular for their superior image reconstruction abilities. These tokenizers have enabled the advent of high-performance, scalable token-based generative models [85, 56, 87, 6]. Early token-based models were mainly autoregressive, generating images one token at a time [48, 13, 11, 85]. In contrast, non-autoregressive transformers (NATs)[7, 33, 6, 53] generate multiple tokens simultaneously, speeding up the process while maintaining high image quality. Recently, visual autoregressive models[70] introduced a next-scale prediction strategy, also demonstrating their promise in image synthesis.

---

[3]We illustrate with 4×4 tokens for simplicity; the actual token map size may be 16×16 or larger.

**Efficient image synthesis** has witnessed significant progress recently. Though the efficiency issue is relatively less explored in token-based image synthesis, it has been extensively studied in diffusion-based models. This includes advanced samplers [40, 41, 37], distillation methods [62, 83], quantization and compression techniques [92, 88, 91], and efforts to reduce redundant computation [42, 79, 1]. The last approach bears some resemblance to our computation reuse mechanism in Sec. 4.2, but with notable differences. Firstly, the subjects of research differ: we focus on NAT models. This focus introduces unique properties, *e.g.*, NATs incrementally decode new tokens at specific spatial locations, resulting in feature maps that are only significantly updated in those areas during generation. This contrasts with diffusion models, where feature map similarity between adjacent steps does not follow such predictable spatial patterns; instead, some layers show high overall similarity within a certain range of timesteps, while others may not. Secondly, these characteristic differences lead to distinct methodologies. Diffusion models typically require manually fine-tuned, and sometimes layer-specific caching schedules [42] to reuse previously computed features. This process can be labor-intensive and may struggle with generalization. In contrast, our method prioritizes model computation on newly decoded tokens in NATs and reuses the final representations of previously computed tokens without manually fine-tuned caching schedules.

**Masked image modeling** (MIM) methods like MAE [29] are widely used for *learning image representations* by predicting missing patches, with the encoder processing visible tokens and the decoder attending to both visible and masked tokens for reconstruction. CrossMAE [15] extends this by adopting a more disentangled architecture for handling both token types separately. In contrast, our work focuses on *image generation*, applying masked image modeling in discrete image token space, where token prediction and reconstruction are required at every step. This introduces key differences, such as SC-Attention and computation reuse mechanisms (see Sec. 4) which are not explored in these MIM approaches.

**Non-autoregressive Transformers (NATs)** originated in machine translation for their fast inference capabilities [19, 20]. Recently, they have been adapted for image synthesis, enabling efficient high-quality image generation as evidenced by various studies [7, 33, 35, 6, 53, 86]. MaskGIT [7] was the first to show NAT's effectiveness on ImageNet. This approach has been expanded for text-to-image generation, scaling up to 3B parameters in Muse [6] and achieving outstanding performance. Token-critic [33] and MAGE [35] enhance NATs further: Token-critic uses an auxiliary model for guided sampling, while MAGE integrates representation learning with image synthesis using NATs. Recent studies [46, 47] have also explored techniques for further improving the training and inference process of NATs. In contrast to these works, we aim to better understand the mechanisms behind NATs' effectiveness, uncovering findings that naturally lead to a more efficient and effective design for NAT models.

## 3   Preliminaries of Non-autoregressive Transformers (NATs)

In this section, we provide an overview of Non-Autoregressive Transformers (NATs) [7, 6, 35] for image generation. NATs operate with a pre-trained VQ-Autoencoder [71, 57, 13], which maps images to discrete visual tokens and reconstructs images from these tokens. The VQ-Autoencoder consists of three components: an encoder $\mathcal{E}^{\mathrm{VQ}}$, a quantizer $\mathcal{Q}$ with a learnable codebook $e$, and a decoder $\mathcal{D}^{\mathrm{VQ}}$. The encoder and quantizer transform an image into a sequence of visual tokens:

$$\boldsymbol{v} = \mathcal{Q}(\mathcal{E}^{\mathrm{VQ}}(\boldsymbol{x})), \tag{1}$$

where $\boldsymbol{v} = [v_i]_{i=1:N}$ is the sequence of visual tokens, and $N$ is the sequence length. Each token $v_i$ corresponds to a specific entry in the VQ-Autoencoder codebook. The above process is known as *tokenization*. After tokenization, NATs learn to generate visual tokens in the latent VQ space.

During training, NATs optimize the masked language modeling (MLM) objective [9]. Specifically, a random subset of tokens is replaced with a special `[MASK]` token, and the model is trained to predict the original tokens based on the unmasked ones. Formally, let $\boldsymbol{M}$ be the mask vector, where $m_i = 1$ indicates the $i$-th token is masked. The training objective minimizes the negative log-likelihood of the masked tokens:

$$L_{MLM} = - \sum_{i \in [1,N], m_i = 1} \log p(v_i | \boldsymbol{v}_{\overline{M}}), \tag{2}$$

where $p(v_i | \boldsymbol{v}_{\overline{M}})$ is the predicted probability of token $v_i$ given the unmasked tokens $\boldsymbol{v}_{\overline{M}}$.

To generate images, NATs follow an iterative decoding strategy [7]. Starting with a fully masked token map, the model predicts all masked positions and samples a portion of the most confident predictions to replace the mask tokens in each iteration. The number of masked tokens to be replaced follows a cosine function, with fewer tokens replaced in the early iterations and more tokens replaced in later iterations. The finally decoded token sequence $\hat{v}$ is then decoded into an image by the VQ-Autoencoder decoder:

$$\hat{x} = \mathcal{D}^{\text{VQ}}(\hat{v}). \tag{3}$$

Due to space limitations, we refer readers to [7] for more details.

# 4 EfficientNAT (ENAT)

In this section, we design several analytical experiments (details in Appendix A.1) to advance the understanding of the mechanisms behind the effectiveness of NATs, aiming to accordingly improve the design of NAT models. Specifically, we uncover the critical spatial and temporal interaction patterns that naturally emerge within NATs under the goal of image generation. Inspired by our findings, we further propose to gradually re-design NATs towards maximally exploiting these characteristics.

## 4.1 Spatial Level Interaction

**Motivation: an ablation study.** A notable characteristic of NATs is the concurrent processing and interaction (through attention layers) of visible ([V]) and [MASK] ([M]) tokens when inferring the unknown image content. To better understand this mechanism, we consider an ablation study on four types of spatial interactions: a) [M] to [V] attention, b) [V] to [M] attention, c) [V] to [V] attention, and d) [M] to [M] attention. We find these four types of spatial interactions have significantly different impacts on the generation performance. As shown in Figure 2, the most important spatial interaction is the [M] to [V] attention (*i.e.*, [V]→[M] information propagation), without which the model is unable to converge at all. Moreover, both [M] to [M] and [V] to [V] attentions (*i.e.*, self-attention within the representation-extraction processes of visible and [MASK] tokens, respectively) moderately improve the model. The most intriguing fact is that removing the [V] to [M] attention (*i.e.*, [M]→[V] information propagation) only marginally hurts the model's performance.

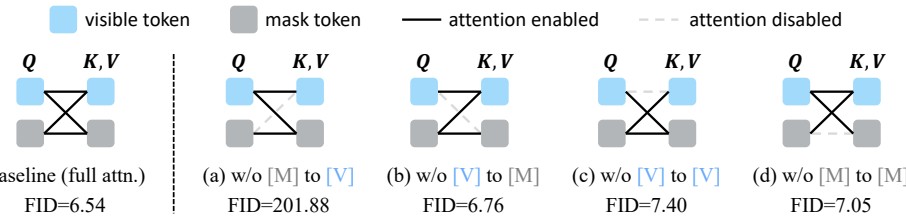

Figure 2: **An ablation study on four types of spatial interactions.** The essential spatial interaction is the [M] to [V] attention. In contrast, the [V] to [M] attention only marginally affects the model.

This imbalanced importance of four spatial interactions highlights the distinct roles of visible and [MASK] tokens. Specifically, the processing of the visible tokens primarily establishes certain internal representations based on the currently available and reliable information, and propagates them to the [MASK] tokens. In fact, their corresponding deep representations can be built mainly on top of themselves. In contrast, [MASK] tokens progressively gather information from visible tokens to predict the proper token values corresponding to the unknown parts of the images. In other words, *NATs naturally separate the role of visible and mask tokens when learning to generate images effectively, even though the two types of tokens are designed to be processed equally in NAT models.*

This phenomenon raises an intriguing question: can we **improve NATs by explicitly encouraging the naturally emergent spatial-level token-interaction patterns**? Actually, this idea is feasible. For example, we can consider a disentangled architecture that explicitly differentiates the roles of visible and [MASK] tokens. As shown in Figure 3b, we may process visible tokens *independently* of [MASK] tokens, with the sole purpose of encoding the current visible and reliable information. In contrast, the computation allocated to [MASK] tokens may only focus on predicting unknown image contents correctly with

Table 1: Effectiveness of disentangled architecture.

| Arch. | GFLOPs | FID↓ |
|---|---|---|
| Baseline | 39.6 | 6.54 |
| Disentangled | 40.2 | **5.50** |

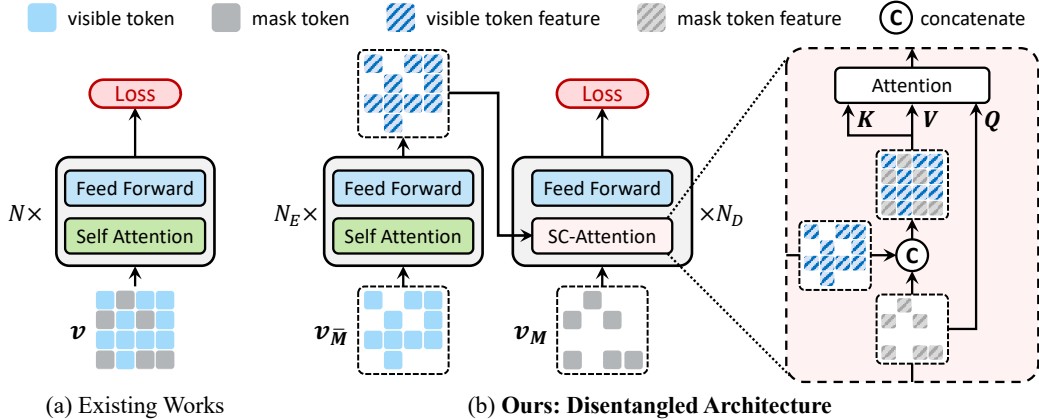

(a) Existing Works      (b) **Ours: Disentangled Architecture**

Figure 3: (a) **Existing works of NATs** process visible and [MASK] tokens equivalently. (b) **Our disentangled architecture** independently encodes visible tokens and integrates their fully contextualized features into the [MASK] token decoding process. $M$ is the indicator of [MASK] tokens while $\bar{M}$ is the indicator of visible tokens. The SC-Attention concatenates the visible and mask token features to produce keys and values, providing a complete context for the mask token decoding.

the help of fully contextualized visible token representations. Such a disentangled architecture significantly improves the performance of NATs at a similar computational cost (see Table 1 for evidence). Inspired by [2], we efficiently integrate the encoded visible tokens into the decoding process of [MASK] tokens with a tailored SC (SelfCross)-attention mechanism (see Figure 3b). The SC-attention simultaneously handles the interactions within [MASK] tokens and the interactions between [MASK] tokens and visible tokens, and it outperforms other possible designs like stacking self-attention and cross-attention layers alternately (see Table 6b).

Moreover, further explorations of our disentangled architecture yield an interesting finding: **prioritizing visible tokens results in an enhanced efficiency**. As shown in Table 2, the paradigm of equal computation allocation across all tokens derived from existing NATs may be far from optimal. Instead, allocating more computation to visible tokens yields notably better performance without sacrificing efficiency, while the computation on masked tokens can be reduced to only a single layer. This observation further underscores the importance of our proposed disentangled paradigm of processing visible tokens from masked ones in enabling advanced network architecture design.

Table 2: Effects of prioritizing visible tokens. $N_E$, $N_D$: encoder/decoder layers (for visible/[MASK] tokens). Network width is slightly adjusted to make GFLOPs approximately unchanged.

| $N_E$ | $N_D$ | GFLOPs | FID↓ |
|---|---|---|---|
| 8 | 8 | 40.2 | 5.50 |
| 12 | 4 | 38.2 | 4.98 |
| 15 | 1 | 39.8 | **4.78** |

## 4.2 Temporal Level Interaction

**Feature similarity across generation steps.** Another critical characteristic of NATs is their incremental revelation of unknown parts of the image upon previous steps. Beyond this straightforward procedure of progressive generation, here we are interested in whether there exist some interpretable temporal interaction patterns in NATs' behaviors. For instance, how do a NAT's computation results at the current step relate to those at the previous step? To investigate this, we conduct a similarity analysis of NATs' output features between two adjacent generation steps.

In Figure 5a, we randomly select two generated samples in NATs and visualize their token feature similarity at two adjacent steps (steps 2 & 3 and steps 6 & 7). We compare token-wise similarity and adopt cosine similarity as the metric: $\text{Sim}(\boldsymbol{z}^{(t-1)}, \boldsymbol{z}^{(t)})_{ij} = \frac{\boldsymbol{z}_{ij}^{(t-1)} \cdot \boldsymbol{z}_{ij}^{(t)}}{\|\boldsymbol{z}_{ij}^{(t-1)}\| \|\boldsymbol{z}_{ij}^{(t)}\|}$, where $\boldsymbol{z}_{ij}^{(t)}$ denotes the feature of the token at position $(i, j)$ and timestep $t$. The similarity map exhibits a highly polarized pattern: token representations undergo drastic changes at some "critical positions", while other positions remain highly similar between adjacent steps. When comparing with the positions of newly decoded tokens, we find that these "critical positions" correspond precisely to where the newly decoded tokens are located. In other words, *the major significance of each time step lies in updating*

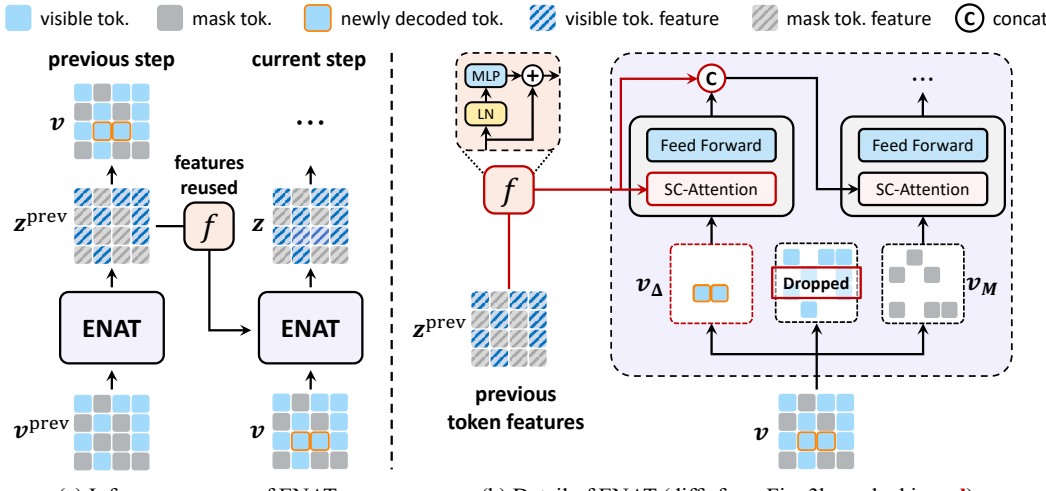

(a) Inference process of ENAT   (b) Detail of ENAT (diffs from Fig. 3b marked in **red**)

Figure 4: **Overview of ENAT.** Based on the disentangled architecture in Fig. 3b, we further propose to only encode the critical (*i.e.*, newly decoded) tokens and maximally reuse previously extracted features to supplement necessary information. $\Delta$ is the indicator of newly decoded tokens. Only one transformer block is illustrated for simplicity.

*the representations of newly decoded tokens, while the computation for the remaining majority of tokens is generally repetitive.* In Figure 5b, we plot the average token similarity over 50,000 generated samples in each pair of adjacent steps ($t = 1 \rightarrow 2$, $t = 2 \rightarrow 3$, ..., $t = 7 \rightarrow 8$). The results show that this temporal interaction pattern remains consistent for different timesteps/samples.

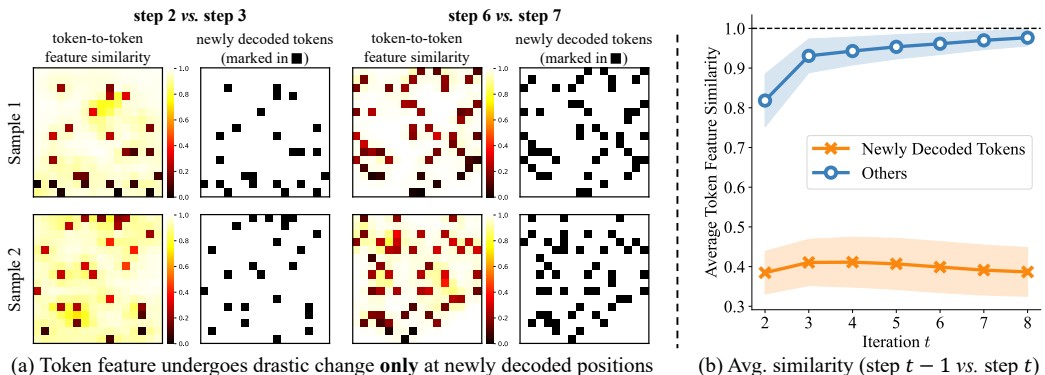

(a) Token feature undergoes drastic change **only** at newly decoded positions   (b) Avg. similarity (step $t - 1$ *vs.* step $t$)

Figure 5: **Feature similarity analysis.** (a) We randomly choose two samples and visualize the token-to-token feature similarity between adjacent steps (2 & 3 and 6 & 7), with the positions of newly decoded tokens visualized on the right. (b) The token feature similarity averaged over 50,000 generated samples in each pair of adjacent steps ($t = 1 \rightarrow 2$, $t = 2 \rightarrow 3$, ..., $t = 7 \rightarrow 8$).

**Computation reuse.** Driven by these observations, our key insight is: *during the generation of NATs, not all tokens need to be re-computed from scratch at each step.* Instead, only the newly decoded tokens need to be re-encoded to inject new knowledge about the image, while the previously encoded information can be maximally reused to supplement necessary details.

To implement this idea, we slightly modify the inference process upon our disentangled architecture (Sec. 4.1) by only encoding the newly decoded tokens at each step, while integrating the previously computed features to assist the current step's decoding:

$$\text{without reuse (Fig. 3b)}: \quad z = \text{Forward}(v_{\bar{M}}, v_M) \quad (4)$$

$$\textbf{with reuse (Fig. 4b)}: \quad z = \text{Forward}(v_{\Delta}, v_M, f(z^{\text{prev}})) \quad (5)$$

where $z^{\text{prev}}$ is the feature computed on the previous step, which is projected by a light-weight projection module $f(\cdot)$, and $v_{\Delta}$ denotes the newly decoded tokens. We adopt the SC-Attention mechanism (Sec. 4.1) to integrate the previous features into the current step's decoding process. At the end of the encoder, we simply concatenate the projected previous features with the computed features on the newly decoded tokens, and feed them into the decoder. In this way, the previously computed features are maximally reused to supplement both the encoding and decoding process of the current step, significantly reducing the computation cost and accelerating the generation process. The detailed inference process is illustrated in Figure 4b.

To equip the model with the ability to utilize previously computed features, we introduce minor modifications to the training process by alternating between normal forward mode and a "reuse forward mode" during training (each with 50% probability).

More specifically, the "reuse mode forward" during training is achieved through the following steps:

1. **Masking Tokens**: Given the current input token map $\mathbf{v}$, we mask a random subset of visible tokens in $\mathbf{v}$ to create $\mathbf{v}^{\text{prev}}$.
2. **Feature Extraction**: Feed $\mathbf{v}^{\text{prev}}$ into the NAT model to obtain its features $\mathbf{z}^{\text{prev}}$.
3. **Forward Pass and Loss Computation**: Use Eq. (5) to forward the current input token map $\mathbf{v}$ along with $\mathbf{z}^{\text{prev}}$ obtained in Step 2, and compute the loss. In practice, a stop gradient operation is applied before feeding $\mathbf{z}^{\text{prev}}$ into Eq. (5).

At other times, we use the original forward mode without incorporating previous features: Eq. (4), where the previous features are empty and the SC-Attention on the left of Figure 4b naturally reduces to the original self-attention mechanism. The reuse mechanism significantly accelerates the generation process, as shown in Table 6a. Additionally, we find in practice that only feeding the visible token feature of the previous step is sufficient and achieves better efficiency, as shown in Table 6d.

## 5 Experiments

**Setups.** Following [7, 35, 6], we utilize a pretrained VQGAN [13] with a codebook of size 1024 for image and visual token conversion. We employ three NAT models: ENAT-S (15 encoder layers, 1 decoder layer, 366 embedding dimensions, primarily for ablations), ENAT-B (15 encoder layers, 1 decoder layer, 768 embedding dimensions), and ENAT-L (22 encoder layers, 2 decoder layers, 1024 embedding dimensions). For class-conditional generation, we use adaptive layer normalization [80, 49] for conditioning. For text-to-image generation, we concatenate text embeddings with visual tokens for conditioning. Our training configurations follow [3] with minor adjustments to batch sizes and learning rates to accommodate different model sizes. For system-level comparisons in Sec. 5.1, we measure the TFLOPs of the entire generation process (including the decoder part for latent space generation models) to ensure fair comparisons.[4] All our experiments are conducted with $8 \times$ A100 80G GPUs. We generally follow the approach described in [3] with minor modifications. More details on the training and inference setups, and the choice of our baselines can be found in Appendix A.2.

### 5.1 Main Results

**Class-conditional generation on ImageNet 256×256 and 512×512.** In Table 3, we compare our approach with other generative models on ImageNet 256×256. Our ENAT achieves superior performance with significantly lower computational cost. For instance, our ENAT-B model, despite having an extremely low inference cost, attains competitive FID scores of 3.53 in 8 steps. With a slightly increased computational budget, our ENAT-L model achieves a FID of 2.79 with only 0.3 TFLOPs, surpassing leading models with substantially less computational effort. For example, compared to the most performant baseline, *i.e.*, U-ViT-H [3], our ENAT-L model achieves a lower FID score (2.79 *vs.* 3.37) while requiring **8×** lower computational cost (0.3 TFLOPs *vs.* 2.4 TFLOPs). We further evaluate our ENAT on ImageNet 512×512 in Table 4. Our ENAT-L model also achieves a superior FID of 4.00 with only 1.3 TFLOPs, outperforming leading models with much lower inference cost. Qualitative results of our method are presented in Figure 7 and Appendix B.

---

[4]This differs from the GFLOPs reported in our ablation studies in Tabs. 1, 2, 6, where VQ-decoder costs are excluded to better compare the efficiency of different NAT designs.

Table 3: **Results on ImageNet 256×256** . TFLOPs quantify the total computational cost for generating a single image. For DPM-Solver [40] augmented diffusion models ($^\dagger$), we follow [40] to tune configurations and report the lowest FID. Diff: diffusion, AR: autoregressive.

| Method | Type | #Params | Steps | TFLOPs↓ | FID↓ | IS↑ |
|---|---|---|---|---|---|---|
| BigGAN-deep [4] (ICLR'19) | GAN | - | 1 | - | 6.95 | 171.4 |
| StyleGAN-XL [63] (SIGGRAPH'22) | GAN | - | 1 | 1.5 | 2.30 | 265.1 |
| VQVAE-2 [57] (NeurIPS'19) | AR | 13.5B | 5120 | - | 31.1 | ∼ 45 |
| VQGAN [13] (CVPR'21) | AR | 1.4B | 256 | - | 15.78 | 78.3 |
| ADM-G [10] (NeurIPS'21) | Diff. | 554M | 250 | 334 | 4.59 | 186.7 |
| LDM [59] (CVPR'22) | Diff. | 400M | 250 | 52.3 | 3.60 | 247.7 |
| LDM$^\dagger$ [59] (CVPR'22) | Diff. | 400M | 4 | 1.2 | 11.74 | - |
| | | | 8 | 2.0 | 4.56 | 262.9 |
| U-ViT-H$^\dagger$ [3] (CVPR'23) | Diff. | 501M | 4 | 1.4 | 8.45 | - |
| | | | 8 | 2.4 | 3.37 | 235.9 |
| DiT-XL$^\dagger$ [49] (ICCV'23) | Diff. | 675M | 4 | 1.3 | 9.71 | - |
| | | | 8 | 2.2 | 5.18 | 213.0 |
| MDT-XL$^\dagger$ [16] (ICCV'23) | Diff. | 676M | 4 | 1.3 | 11.36 | - |
| | | | 8 | 2.2 | 4.00 | - |
| USF [38] (ICLR'24) | Diff. | 554M | 8 | 10.7 | 9.72 | - |
| MaskGIT [7] (CVPR'22) | NAT | 227M | 12 | 1.22 | 4.92 | - |
| Token-Critic [33] (ECCV'22) | NAT | 422M | 36 | 1.9 | 4.69 | 174.5 |
| Draft-and-revise [32] (NeurIPS'22) | NAT | 1.4B | 72 | - | 3.41 | 224.6 |
| MAGE [35] (CVPR'23) | NAT | 230M | 20 | 1.0 | 6.93 | - |
| MaskGIT-FSQ [43] (ICLR'24) | NAT | 225M | 12 | 0.8 | 4.53 | - |
| AdaNAT [47] (ECCV'24) | NAT | 206M | 8 | 0.9 | 2.86 | 265.4 |
| **ENAT-B** | NAT | 219M | 4 | 0.1 | 5.86 | - |
| | | | 8 | 0.2 | 3.53 | 302.4 |
| **ENAT-L** | NAT | 574M | 4 | 0.2 | 4.13 | - |
| | | | 8 | 0.3 | **2.79** | **326.7** |

Table 4: **Results on ImageNet 512×512**. $^\dagger$: DPM-Solver [40] augmented diffusion models.

| Method | Type | #Params | Steps | TFLOPs↓ | FID↓ | IS↑ |
|---|---|---|---|---|---|---|
| VQGAN [13] (CVPR'21) | AR | 227M | 1024 | - | 26.52 | 66.8 |
| ADM-G [10] (NeurIPS'21) | Diff. | 559M | 250 | 579 | 7.72 | 172.7 |
| U-ViT-H$^\dagger$ [3] (CVPR'23) | Diff. | 501M | 8 | 3.4 | 4.60 | **286.8** |
| DiT-XL$^\dagger$ [49] (ICCV'23) | Diff. | 675M | 8 | 9.6 | 5.44 | 275.0 |
| MaskGIT [7] (CVPR'22) | NAT | 227M | 12 | 3.3 | 7.32 | 156.0 |
| MaskGIT-RS [7] (CVPR'22) | NAT | 227M | 12 | 13.1 | 4.46 | - |
| Token-Critic [33] (ECCV'22) | NAT | 422M | 36 | 7.6 | 6.80 | 182.1 |
| Token-Critic-RS [33] (ECCV'22) | NAT | 422M | 36 | 34.8 | 4.03 | - |
| **ENAT-L** | NAT | 574M | 8 | **1.3** | **4.00** | 285.7 |

**Text-to-image generation on MS-COCO.** We further assess the efficacy of ENAT for text-to-image generation on MS-COCO [36]. Table 5 shows that ENAT-B surpasses competing baselines with just 0.3 TFLOPs, achieving a FID score of 6.82. Compared to the competitive diffusion model U-ViT [3] with a fast sampler [41], ENAT-B requires similar computational resources to its 4-step variant while significantly outperforming it (6.82 *vs.* 16.20), and it also surpasses the 8-step sampling results of U-ViT with lower computational costs.

Table 5: **Results on MS-COCO**; all models are trained and evaluated on MS-COCO. $^\dagger$: DPM-Solver [40] augmented diffusion models.

| Method | #Params | Steps | TFLOPs↓ | FID↓ |
|---|---|---|---|---|
| VQ-Diffusion [21] | 370M | 100 | - | 13.86 |
| Frido [14] | 512M | 200 | - | 8.97 |
| U-Net$^\dagger$ [3] | 53M | 50 | - | 7.32 |
| U-ViT$^\dagger$ [3] | 44M | 4 | 0.4 | 16.20 |
| | | 8 | 0.5 | 6.92 |
| **ENAT-B** | 116M | 8 | **0.3** | **6.82** |

**Practical efficiency.** We provide more comprehensive comparisons of the trade-off between generation quality and computational cost in Figure 6. Both theoretical TFLOPs and the practical GPU/CPU latency for generating an image are reported. Our results show that ENAT consistently outperforms other baselines in terms of both generation quality and computational cost.

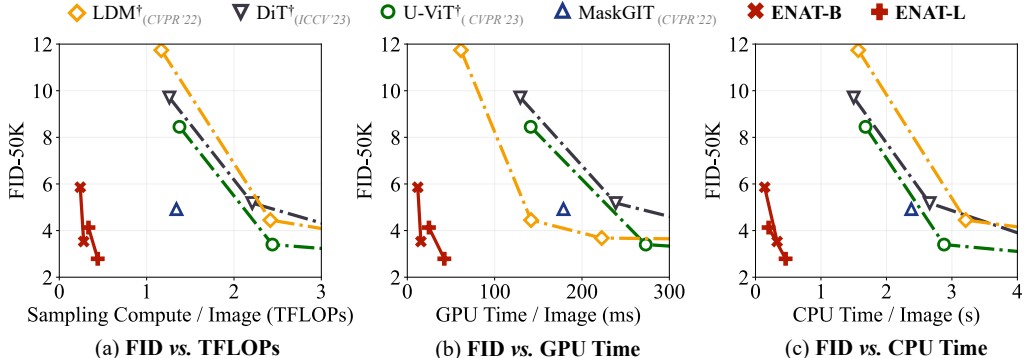

Figure 6: **Practical efficiency of ENAT** . As a reference, we also plot the TFLOPs for generating a single image in (a). GPU time is measured on an A100 GPU with batch size 50. CPU time is measured on Xeon 8358 CPU with batch size 1. [†]: DPM-Solver [40] augmented diffusion models.

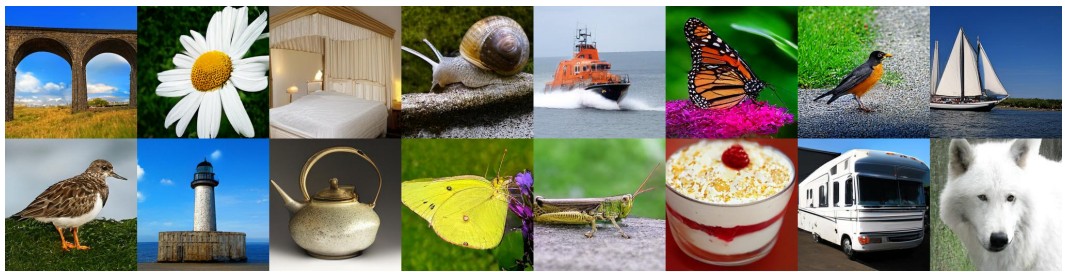

Figure 7: **Selected samples of ENAT-L** with 8 generation steps on ImageNet 256×256.

## 5.2 Ablation Studies

In this section, we present additional ablation studies on Imagenet 256×256 to validate the effectiveness of our proposed mechanisms. We use ENAT-S with 8 generation steps as our default setting, and report the FID score as well as the computational cost in GFLOPs for each NAT model.

**Main ablation.** Disentangled architecture and computation reuse are the two fundamental mechanisms in ENAT. The former separates the processing of visible and [MASK] tokens, and prioritizes computation on visible ones, while the latter eliminates repetitive processing of non-critical tokens. In Table 6a, we demonstrate the effectiveness of these two mechanisms. The results show that the disentangled architecture significantly improves NAT's performance, with a 1.76 improvement in FID score at a similar computational cost. Computation reuse, on the other hand, significantly reduces computational cost (1.8× fewer GFLOPs) while preserving most of the gains from disentanglement.

**Effectiveness of SC-Attention.** The SC-Attention mechanism adopted in our work serves dual roles: handling interactions of input tokens while simultaneously incorporating necessary additional information. Theoretically, the same functionality can be achieved with a stack of one self-attention layer and one cross-attention layer. However, as shown in Table 6b, SC-Attention outperforms the stack of self-attention and cross-attention layers with a lower FID (4.97 *vs.* 5.85) and a lower computational cost (22.6 *vs.* 25.0), demonstrating its effectiveness in our ENAT model.

**Effectiveness of reuse projection module.** In our computation reuse mechanism, a lightweight reuse projection module first processes the previous feature before integrating it into the current generation step. As shown in Table 6c, this design is highly important to our reuse mechanism. Without this module, the FID is 5.96, which is much worse than the 4.78 FID achieved without reuse. An intuitive explanation is that the reuse projection module learns the minimal necessary updates for the features of non-crucial tokens, preventing them from becoming too stale for more distant subsequent steps.

**Which token features to reuse?** Our basic reuse formulation integrates all previous token features into the current step. However, as shown in Table 6d, reusing only visible token features is equally effective while being much more efficient. As discussed in Section 4.1, encoding visible tokens

Table 6: **Ablation studies on ImageNet 256×256**. We use ENAT-S with 8 generation steps as our default setting, which is marked in gray . We report FID-50K following [3, 49] and total GFLOPs for each NAT model throughout the generation process.

(a) **Main ablation.** Our disentangled architecture and reuse mechanism significantly improves NATs.

| Disentangle | Reuse | FID↓ | GFLOPs↓ |
|:---:|:---:|:---:|:---:|
| | | 6.54 | 39.6 |
| ✓ | | **4.78** | 39.8 |
| ✓ | ✓ | 4.97 | **22.6** |

(b) **SC-Attention** outperforms alternately stacking self&cross attention layers with fewer GFLOPs.

| Attn. Type | FID↓ | GFLOPs↓ |
|:---:|:---:|:---:|
| SC | **4.97** | **22.6** |
| self + cross | 5.85 | 25.0 |

(c) **Reuse projection** is lightweight yet critical for maintaining performance.

| Proj. | FID↓ | GFLOPs↓ |
|:---:|:---:|:---:|
| ✓ | **4.97** | 22.6 |
| ✗ | 5.96 | **20.8** |

(d) **Which token features to reuse?** Reusing only visible token features of previous step is sufficient and much more efficienct.

| Prev. Token Features | FID↓ | GFLOPs↓ |
|:---:|:---:|:---:|
| all | **4.95** | 37.5 |
| visible only | 4.97 | **22.6** |

(e) **Which layer of feature to reuse?** Reusing last layer prev. features for all current layers is better than reusing in a layer-to-layer correspondence manner.

| Prev. Feature Pos. | FID↓ | GFLOPs |
|:---:|:---:|:---:|
| last layer | **4.97** | **22.6** |
| layer-to-layer | 5.77 | 36.1 |

is most critical for NAT, and thus our ENAT model focuses most computation on these tokens. Therefore, using only visible token features suffices to provide the necessary information for reuse.

**Which layer of feature to reuse?** We compared reusing the last layer's features from the previous step with reusing features in a layer-by-layer manner, where the $i$-th layer of the current step reuses the features from the $i$-th layer of the previous step. As shown in Table 6e, reusing features from the last layer of the previous step outperforms the layer-by-layer approach, achieving a lower FID of 4.97. Additionally, it requires fewer GFLOPs (22.6 vs. 36.1), as the layer-by-layer approach needs to project features of each previous layer, while the last layer approach only projects once.

## 6 Conclusion

In this paper, we explored the underlying mechanisms of non-autoregressive Transformers (NATs) and uncovered key spatial and temporal token interaction patterns exist within NATs. Our findings highlight that spatially, visible tokens primarily provide information for [MASK] tokens, while temporally, updating the representations of newly decoded tokens is the main focus across generation steps. Driven by these findings, we propose ENAT, a NAT model that explicitly encourages these critical interactions. We spatially disentangle the computations of visible and [MASK] tokens by independently encoding visible tokens and conditioning [MASK] tokens on fully encoded visible tokens. Temporally, we focus computation on newly decoded tokens at each step, while reusing previously computed representations to facilitate decoding. Experiments on ImageNet and MS-COCO demonstrate that ENAT enhances NATs' performance with significantly reduced computational cost.

## Acknowledgements

This work is supported in part by the National Natural Science Foundation of China under Grants 42327901.

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

# A Implementation Details

## A.1 Detailed model configurations.

Here we present the detailed configurations of all our NAT models appeared within this paper in Table 7. We provide the number of encoder layers ($N_E$), decoder layers ($N_D$), the dimension of the hidden states (embed dim.), the number of attention heads (# attn. heads):

Table 7: **Summary of model configurations.** $N_E$: encoder layers (for visible token encoding), $N_D$: decoder layers (for [MASK] token decoding). $^*$: In conventional NAT models, the layers for visible token encoding are shared with the layers for [MASK] token decoding.

| arch. | reuse? | $N_E$ | $N_D$ | embed dim. | # attn. heads |
|---|---|---|---|---|---|
| baseline | ✗ | 8$^*$ | | 288 | 6 |
| disentangled | ✗ | 8 | 8 | 288 | 6 |
| disentangled | ✗ | 12 | 4 | 318 | 6 |
| disentangled | ✗ | 15 | 1 | 366 | 6 |
| ENAT-S | ✓ | 15 | 1 | 366 | 6 |
| ENAT-B | ✓ | 15 | 1 | 768 | 8 |
| ENAT-L | ✓ | 22 | 2 | 1024 | 16 |

## A.2 Details of training and evaluation.

For ImageNet 256×256, we use a batch size of 2048 and a learning rate of 4e-4. For ImageNet 512×512, to manage the increased sequence length, we reduce the batch size to 512 and linearly scale down the learning rate to 1e-4. For MS-COCO, we train for 150k steps instead of the 1000k steps used in [3].

For our ablation studies in Sec. 5.2 and explorative experiments in Sec. 4, we train the models for 300k steps instead of the 500k steps used in [3], while keeping the other settings the same as above.

For data preprocessing, we perform center cropping and resizing to 256×256 for ImageNet 256×256 and MS-COCO, and to 512×512 for ImageNet 512×512. Additionally, we adopt random horizontal flipping as data augmentation, following [3, 49].

Our evaluation on FID follows the same evaluation protocol as [10, 3, 49]. We adopt the pre-computed dataset statistics from [3] and generate 50k samples for ImageNet (30k for MS-COCO) to compute the statistics for the generated samples, using the following formula to calculate FID [30]:

$$\text{FID} = ||\mu_{\text{real}} - \mu_{\text{fake}}||_2^2 + \text{Tr}(\Sigma_{\text{real}} + \Sigma_{\text{fake}} - 2(\Sigma_{\text{real}}\Sigma_{\text{fake}})^{1/2}), \tag{6}$$

where $\mu$ and $\Sigma$ are the mean and covariance of the real and fake samples, respectively. The evaluation on Inception Score (IS) follows the same protocol as [3, 49], using a pre-trained InceptionV3 model [66] to compute the IS.

For the choice of baselines in our work, since ENAT focuses on inference efficiency, we aim to compare ENAT with other models in a lightweight, low-FLOPs scenario. However, while the inference efficiency of generative models is important, it is generally under-explored in the original papers of state-of-the-art diffusion models (e.g., DiT [49], MDT [16]), which mostly focus on enhancing generation performance. The official results of them are primarily obtained with hundreds of inference steps, making direct comparisons with ENAT challenging. For instance, as shown in Fig. 8, the official results of DiT, MDT, etc. all concentrate at the high end of overall inference costs, requiring hundreds of times more computation than ENAT.

Fortunately, there are well-established fast sampling techniques (e.g. DPM-Solver [40]) for accelerating diffusion models, which allows us to reduce their sampling steps and compare them with ENAT in a fairer setting.

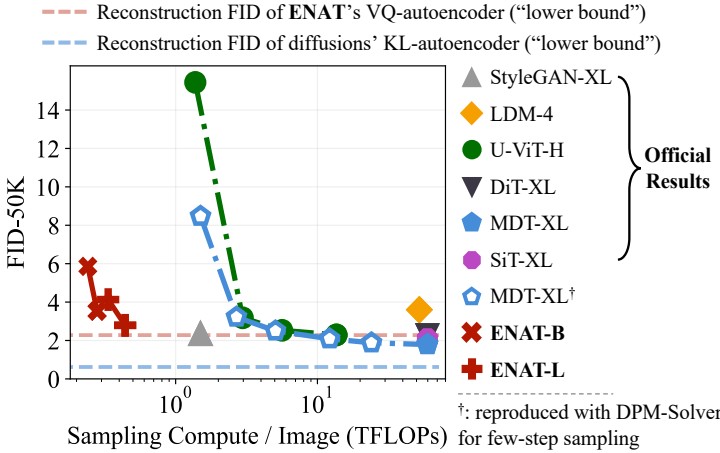

Figure 8: **System-level comparisons on ImageNet 256×256.** All baseline results are sourced from their original papers, except for the few-step MDT results ($^\dagger$).

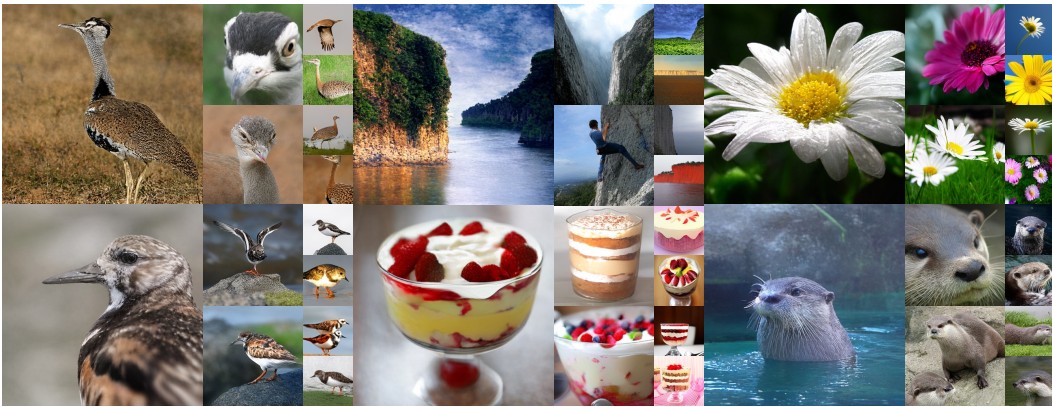

Figure 9: **Selected samples of ENAT-L** with 8 generation steps on ImageNet 256×256 and 512×512.

# B    More Qualitative Results.

Here we present more qualitative results in Figure 9. For each class, the first two columns contain 3 ImageNet 512×512 samples and the last column contains 4 ImageNet 256×256 samples.

# C    Limitations and Future Work

Although our experiments have covered two fundamental types of generative models, namely class-conditional and text-to-image generation, and utilized three datasets, investigating the efficacy of ENAT on more diverse datasets, such as the widely used CelebA [39] and LSUN [84], and exploring additional generation types like unconditional generation, constitute valuable directions for future research. Moreover, scalability, both in terms of model size and dataset volume, is a crucial capability for current generative models. Our largest model scales up to approximately 0.6 billion parameters, and our experiments utilized datasets with a maximum size of 1.2 million images (ImageNet dataset). Evaluating the performance of ENAT on even larger-scale datasets, such as LAION-5B [64], and further scaling the model to surpass 1 billion parameters, could provide deeper insights into its scalability and robustness.

To further enhance the applicability and efficiency of non-autoregressive Transformers, integrating other adaptive inference methods [76, 75, 26, 95] and learning techniques [67, 82, 77] will be essential. For instance, methods like dynamic neural network [74, 27, 89, 93, 94, 52] and resolution-adaptive models [81] offer promising pathways to explore. Additionally, examining ENAT across

Table 8: **Licenses for existing assets.**

| dataset / code | source | license |
|---|---|---|
| MS-COCO | [36] | New BSD License |
| ImageNet | [60] | Custom (research, non-commercial) |
| MaskGIT | [7] | Apache-2.0 license |
| U-ViT | [3] | MIT license |

diverse tasks and domains [51, 50, 28] and leveraging advances in model training and inference techniques [45, 22, 25, 23, 78, 24] can strengthen its performance and expand its scope.

## D Broader Impacts

On the positive side, the proposed EfficientNAT (ENAT) models significantly reduce computational costs, making advanced visual generation technology more accessible. This democratization can benefit diverse sectors, including education, healthcare, and creative industries. However, as with any AI-generated content technology, there are potential ethical considerations such as creating misleading content or spreading misinformation. Additionally, like other data-driven approaches, the model may inadvertently reinforce biases present in the training data. Possible mitigation strategies for these concerns include developing robust detection methods for generated content, promoting transparency in AI-generated content, and ensuring diverse and representative training data.

## E Licenses

The Table 8 outlines the assets used in our work, their sources and licenses. Our models, data and code will be open-sourced under the MIT License upon paper acceptance.

