# OpenReview forum: "ENAT: Rethinking Spatial-temporal Interactions in Token-based Image Synthesis"
_NeurIPS.cc/2024/Conference — NeurIPS 2024 poster_

### Official Review · Reviewer_bqgC · 2024-07-08

**Soundness:** 3
**Presentation:** 2
**Contribution:** 2
**Rating:** 6
**Confidence:** 4

**Summary:**

The paper analyzes the information flow in Non-autoregressive Transformers image generation methods (NAT) to improve their efficiency.
In particular, they find that in a forward pass, having the observed tokens attend to masked token is unnecessary and proposes to change the architecture accordingly.
Moreover, they show that in the iterative steps, many token features don't change across steps and thus don't need to be recomputed.
By taking into account these two observations, their proposed ENAT design drastically cut down the number of TFLOPs their architecture requires while also improving image generation quality compared to the baselines.

**Strengths:**

- The paper tackles the promising topic of Non-autoregressive Transformers for Image Generation.
- It proposes two well-motivated and clearly exposed efficiency improvements backed by an empirical analysis and that also leads to performance improvement on the benchmarks. The findings with temporal redundancy with a projection is a fresh new perspective.
- The paper has a complete set of empirical studies, including on MS-COCO generation and ablation of their modules.

**Weaknesses:**

- The efficiency of masked image transformer is very studied in the adjacent self-supervised masked auto-encoders (MAE) litterature, with whom NAT shares both architectures and base training algorithm, the main differences being different mask schedules and the classification loss in NAT instead of regressions (and even so, BeiT-v2 [A] uses a classification loss as well) .
In this adjacent literature, the seminal MAE paper [B] already had differentiated processes for observed and masked tokens: the masked tokens were introduced later, in the decoder, to avoid unnecessary processing in the encoder part. Recently, CrossMAE [C] replaced the self-attention from the MAE by a cross-attention CA(q=M, kv=V) making the architecture even closer to the proposed ENAT. The main difference is that ENAT uses CA(q=M, kv=concat(M,V)), reintroducing self-attention between masked tokens.
I believe these considerations are very relevant to the discussion and should be included in a prominent position in the paper.
- The SelfCrossAttention module is a minor variation upon standard attention modules that can also be found in the details of prior work, for instance in the Perceiver Resampler module of Flamingo [D].
- The choices of baseline models and scores seem arbitrary and are not motivated. If all prominent families of generative models were to be represented, it is missing StyleGAN-XL [E]. Other very strong contenders such as Simple Diffusion [F], SiT [G], or MDT [H], also could be included, I'm not sure why they are not considered in the tables. For DiT-XL [32], I'm not sure where the reported scores are from. I might have missed them, but I can't seem find these values in the ICCV camera ready and suppmat, or any of the arxiv version.
- The paper shows very few qualitative samples, and only on ImageNet 256x256. Considering the brittleness of image generation evaluation benchmarks, it's good to be able to check what type of images the model produces, especially since different methods can struggle with different aspects.


---

[A] BEiT v2: Masked Image Modeling with Vector-Quantized Visual Tokenizers. Peng et al. arXiv 2022.

[B] Masked Autoencoders Are Scalable Vision Learners. He et al. CVPR 2022.

[C] CrossMAE: Rethinking Patch Dependence for Masked Autoencoders. Fu et al. arXiv preprint Jan 2024.

[D] Flamingo: a Visual Language Model for Few-Shot Learning. Alayrac et al. NeurIPS 2022.

[E] StyleGAN-XL: Scaling StyleGAN to Large Diverse Datasets. Sauer et al. SIGGRAPH 2022.

[F] simple diffusion: End-to-end diffusion for high resolution images. Hoogeboom et al. ICML 2023.

[G] SiT: Exploring Flow and Diffusion-based Generative Models with Scalable Interpolant Transformers. Ma et al. arXiv preprint Jan 2024.

[H] Masked Diffusion Transformer. Gao et al. ICCV 2023.

**Questions:**

- It is mentionned in A2 that the experiments follow the approaches of [2], that is a Diffusion Transformer. By cross-referencing Table 6.a and Table 3, it seems this is a worse starting point than MaskGIT. Is that correct? Can we expect even better performances by starting from a strong model?
- What motivated the choice of the baselines? Where are the DiT-XL scores from?
- Are the FID computed against training or validation sets of ImageNet?

The submission has a principled approach and offers interesting findings. However, its originality is somewhat dimmed by the missing related work. Clarifications about the choice of baselines are also needed, along with many more qualitative samples, to make the comparisons more transparent. I'd be happy to switch my recommendations to the positive side if the necessary clarifications are made and if the authors demonstrate their willingness to include the requested discussions in the submission.

**Limitations:**

Limitations and societal impacts are adressed adequately in the appendix.

---

> ### Author Rebuttal · Authors · 2024-08-06
>
> > **W1: Relationship with MAE, CrossMAE**
>
> Thanks for your valuable suggestion. We are happy to include more discussions in our revision.
>
> **Firstly**, we kindly note that our approach includes *two* key aspects: a *disentangled architecture* and a *computation reuse* mechanism. To the best of our knowledge, the latter is novel in NAT and highly effective, yet has not been explored in MAE/CrossMAE since they focus on **representation learning**, which does not involve a multi-step process that exists in **generation tasks**.
>
> **Secondly**, while our *disentangled architecture* may appear similar to MAE and CrossMAE, it contains subtle yet important differences due to the distinction between representation learning and generation tasks.
> Specifically, MAE inputs all tokens into the decoder and employs full attention on them ([M] & [V] → [M] & [V]), and CrossMAE retains a single "[V] → [M]" information flow. In contrast, we adopt a "[M] & [V] → [M]" information flow in the decoder attention:
> |**Decoder Attention Type**|**Decoder Attention Information Flow**|**GFLOPs↓**|**FID↓**|
> |-|-|-|-|
> |MAE|[M] & [V] → [M] & [V]|24.5|5.03|
> |CrossMAE|[V] → [M]|21.4|6.31|
> |Ours|[M] & [V] → [M]|22.6|**4.97**|
>
> The results show that directly using MAE-type full attention merely adds computational overhead without improving performance, whereas implementing CrossMAE-type attention degrades performance notably.
>
> This also aligns with our observation in Sec. 4.1 regarding generation tasks:
> 1. Visible tokens primarily provide information. Since the "[V] → [M]" component in our "[M] & [V] → [M]" attention already fulfills the information-providing role of visible tokens, additionally incorporating the visible tokens into the decoder input for full attention (as in MAE) provides no benefit.
>
> 2. The attention between masked tokens cannot be ignored (as opposed to CrossMAE). Intuitively, this allows masked tokens to be aware of each other during decoding, thus preventing incongruencies in the generated results. This further highlights the distinction between a generation-focused approach and a representation learning-focused approach.
>
> **Finally**, we would like to clarify that our contributions are not limited to proposing a new architecture/mechanism.
> We have revealed several important scientific findings and insights in NATs, which are also novel contributions.
> We believe these findings can provide new perspectives for the research community.
> > **W2: Novelty of SelfCrossAttention Module**
>
> Thanks for pointing out the related work [D], and we are happy to update our revision with this reference.
>
> Additionally, we would like to kindly clarify that the SC-attention, as briefly introduced in Section 4.1, is not presented as our main contribution. Our primary contribution lies in uncovering new scientific findings in NAT generation, specifically regarding its spatial and temporal properties, that enhance our understanding and enable a better ENAT design.
>
> > **W3 & Q2: Choices and Scores of Baseline Models**
>
> Thanks for your suggestion. We are happy to add GAN-type models like StyleGAN-XL in our revision, and we have already incorporated it in **Fig. 3** of our global response.
>
> For the choice of baselines in our work, since ENAT focuses on inference efficiency, we aim to compare ENAT with other models in a lightweight, low-FLOPs scenario.
> However, while the inference efficiency of generative models is important, it is generally under-explored in the original papers of state-of-the-art diffusion models (e.g., DiT, MDT, SiT), which mostly focus on enhancing generation performance.
> The official results of them are primarily obtained with hundreds of inference steps, making direct comparisons with ENAT challenging.
> For instance, as shown in **Fig. 3**, the official results of DiT, MDT, and SiT all concentrate at the high end of overall inference costs, requiring hundreds of times more computation than ENAT.
>
> Fortunately, there are well-established fast sampling techniques (e.g. DPM-Solver) for accelerating diffusion models, and we have identified several diffusion models that are both highly competitive and officially supported* by these acceleration techniques.
> This allows us to reduce their sampling steps and compare them with ENAT in a fairer setting.
>
> For the mentioned stronger contenders, we plotted MDT and SiT in **Fig. 3**. Simple diffusion is omitted because its computational cost is neither reported nor open-sourced for measurement.
> The strongest MDT model is also integrated with fast sampler for fewer sampling results.
> Few-step fast sampler results from the original U-ViT paper are also included for reference.
>
> In our revision, we are willing to add more discussions on our baseline results and setups. Stronger baseline models like MDT and SiT will also be included.
>
> ---
> *: U-ViT has DPM-Solver integrated in their Github repo. DiT and LDM are in the Hugging Face `diffusers` library, where DPM-Solver is also officially supported.
> > **W4: More Qualitative Results**
>
> Thanks for your suggestion. In our global response, we have added qualitative comparisons with DiT-XL in **Fig. 4** and included many more qualitative samples for all evaluated datasets in **Fig. 5 & 6**.
> More samples will also be added in our revision.
>
> > **Q1: U-ViT as a Worse Starting Point?**
>
> We would like to clarify that ENAT only follows the *training pipeline* of [2], rather than its diffusion transformer model architecture.
> The pipeline of [2] is relatively standard, and our goals are twofold: **1)** to validate the gains solely attributable to our design under fair conditions, and **2)** to demonstrate that our method does not rely on specific training techniques.
> Exploring more advanced training techniques could potentially lead to further improvements, and we are happy to explore this in the future.
> > **Q3: FID Evaluation Protocol**
>
> We follow the widely adopted OpenAI's evaluation protocol to compute FID against the ImageNet training set.

---

> ### Comment · Reviewer_bqgC · 2024-08-12
>
> Thank you for the detailed answers and important clarifications.
>
> I don't really have major concerns left, and will update my rating accordingly.

---

> > ### Author Response · Authors · 2024-08-13
> > **Thank you for your time and effort**
> >
> > We’re pleased that your concerns have been addressed. Your professional and constructive feedback has helped to clarify and strengthen our work.
> >
> >  We will revise the paper accordingly based on our discussion. Thank you once again for your valuable insights and support!

---

### Official Review · Reviewer_BoDK · 2024-07-14

**Soundness:** 3
**Presentation:** 3
**Contribution:** 3
**Rating:** 5
**Confidence:** 4

**Summary:**

This manuscript reveals NATs' progressive token revelation, with asymmetric intra-step interactions where [MASK] tokens gather info and visible tokens offer it, focusing on key updates amidst repetition across steps. It then presents ENAT, which isolates [MASK]/visible token computations, prioritizes crucial updates, and recycles representations, significantly boosting NAT efficiency and performance on benchmarks like ImageNet and MS-COCO, with code and models to follow acceptance.

**Strengths:**

+ The structure of the paper is well-organized, and the logic of the arguments is clear, making it easy to understand.
+ The motivation is clear and well-founded, leading to compelling results.

**Weaknesses:**

- Regarding the methodology, concerning Figure 5(b), how are the parameters within f(z^pre) (cross-step) trained?
- Regarding scalability, in Table 3, ENAT-L achieves state-of-the-art (SOTA) performance with merely 574M parameters. What happens if larger models are employed? Does it possess a consistent ability to scale up efficiently, or does its optimization primarily exhaustively refine smaller models?
- In Figure 6, it appears that the performance has not yet converged, suggesting that further increases in model size could potentially lead to significant additional performance gains.
- The manuscript lacks sufficient visualized results, especially those that reveal how the method works and why it leads to performance enhancements.

**Questions:**

Please refer to my weakness.

**Limitations:**

Yes, the authors have discussed the limitations of this manuscript in the supplementary material.

---

> ### Author Rebuttal · Authors · 2024-08-06
>
> > **W1: How the Projection Module $f$ is Trained?**
>
> As discussed in line 221-224, we execute the "reuse mode forward" (Eq. (5)) with some probability during training, thus involving the parameters of $f$ in the computation graph. More specifically, the "reuse mode forward" during training is achieved through the following steps:
>   1. **Masking Tokens**: Given the current input token map $\mathbf{v}$, we mask a random subset (*e.g.*, 30%) of visible tokens in $\mathbf{v}$ to create $\mathbf{v}^{\text{prev}}$.
>   2. **Feature Extraction**: Feed $\mathbf{v}^{\text{prev}}$ into the NAT model to obtain its features $\mathbf{z}^{\text{prev}}$.
>   3. **Forward Pass and Loss Computation**: Use Eq. (5) to forward the current input token map $\mathbf{v}$ along with $\mathbf{z}^{\text{prev}}$ obtained in Step 2, and compute the loss. In practice, a stop gradient operation is applied before feeding $\mathbf{z}^{\text{prev}}$ into Eq. (5).
>
> By following Steps 1 and 2, we construct $\textbf{z}^{\text{prev}}$ and feed it into the projection module $f$ in Step 3, thus involving $f$ in the forward propagation. Since all operations are differentiable, gradients can then be effectively back-propagated to update the parameters of $f$.
>
> We will elaborate further in our revision to make this process clear.
>
> > **W2 & W3: Scalability of ENAT**
>
> In line with findings from other token-based models [A, B], ENAT is also scalable with increased model size.
> Due to limitations in time and computational resources, we conducted a preliminary experiment using a 1B parameter version of ENAT, referred to as ENAT-XL (28 layers, 1280 width):
>
> |         | **#Params** | **FID↓** |
> | ------- | ------- | ---- |
> | ENAT-L  | 574M    | 2.79 |
> | ENAT-XL | 1B      | **2.48** |
> | reconstruction FID of Autoencoder    | /       | 2.28 |
>
> As the results show, scaling up ENAT can further improve performance.
>
> An interesting finding is that, the performance ENAT-XL is already approaching the reconstruction FID of our VQ-autoencoder, which measures the quality of images decoded from ground truth visual tokens (thus being an indicator of the “performance upper-bound” for generative models in this VQ space).
>
> Therefore, for even further improvements over ENAT-XL, we may consider simultaneously scaling up the generative model and the VQ-autoencoder, which we are actively working on.
>
> ---
>
> [A] Tian, Keyu, et al. "Visual autoregressive modeling: Scalable image generation via next-scale prediction." arXiv preprint arXiv:2404.02905 (2024).
>
> [B] Chang, Huiwen, et al. "Muse: Text-To-Image Generation via Masked Generative Transformers." International Conference on Machine Learning. PMLR, 2023.
>
> > **W4: More Visualization Results Revealing How the Method Works**
>
> Thanks for your suggestion. We have included additional visualization results in our global response.
>
> Specifically, **Fig. 1** visualizes the intermediate generation results throughout the process.
> The results suggest that ENAT generates superior images as it converges to decent quality results faster, and continues to refine for even better image quality till the end.
>
> To further corroborate this point, we conducted a "one-step decoding" experiment.
> In this experiment, we applied various mask ratios (simulating inputs at different stages of generation) to ground truth image token maps and measured the quality of the generated images using FID for one-step decoded results (i.e., the model directly predicts all masked positions in one forward pass).
>
> |  **Mask Ratio** | 0.2 | 0.4 |0.6 | 0.8 |
> |--------------|-------|-------|--------|--------|
> | MaskGIT (Baseline)       |  3.89 | 8.23 | 19.20 | 48.78 |
> | ENAT (Ours)     |  **3.48** | **6.74** | **16.04** | **45.51** |
> | $\Delta$     |  0.41 | 1.49 | 3.16   | 3.27   |
>
> As the results indicate, ENAT consistently outperforms the standard NAT design. The results for the 0.2 mask ratio setting are also visualized in **Fig. 2**. This demonstrates that ENAT's predictions for masked positions are more reliable at each decoding step, allowing it to converge to high-quality results faster and produce superior images.
>
> Finally, we have included more qualitative comparisons and samples in **Fig. 4, 5, and 6** to further validate the effectiveness of our method.

---

### Official Review · Reviewer_GVb9 · 2024-07-15

**Soundness:** 3
**Presentation:** 3
**Contribution:** 3
**Rating:** 6
**Confidence:** 4

**Summary:**

The paper deals with improving the computational efficiency of image
generation non-autoregressive transformers (NATs) (e.g. MASKGIT). NATs
are usually trained by a fill-in-the-blanks objective and sample by
iteratively predicting missing tokens. The approach is based on
separately encoding observed and missing (masked) tokens, as well as
avoiding repeated computation during sampling iterations. Experimental
results show the effectiveness of the approach, achieving better
performance than the baselines at lower computational cost.

**Strengths:**

- The paper is well written and the concepts are clearly explained and
  illustrated.
- The problem of designing more efficient NATs is an important one in
  the context of generative models dealing with increasingly longer
  sequences (e.g. video).
- The method is simple yet effective, and the paper does a good job of
  justifiying the proposed components small experiments.
- Experimental results convincingly show the proposed approach
  achieves better performance than baselines at a much lower
  computational cost.
- Ablation studies clearly isolate the contribution of the proposed
  components.

**Weaknesses:**

- The idea of only encoding visible tokens was already introduced in
  e.g. MAE, MAGE. Is there a significant difference in the approach
  presented here in that regard?
- In the comparisons, the proposed ENAT uses significantly less sample
  steps than the compared baseline NATs. Getting better performance with
  less sampling steps is obviously a positive thing. But this also
  confounds the comparison in reduced flops due to the
  architecture/procedure. I wonder what is the intuition for this
  approach allowing to significantly reduce the number of sampling
  (iterative refinement) steps?

**Questions:**

Please see weaknesses.

---

> ### Author Rebuttal · Authors · 2024-08-05
>
> >**W1: Relationships with MAE and MAGE**
>
> **Firstly**, we kindly clarify that our approach includes *two* key aspects: a *disentangled architecture* and a *computation reuse* mechanism. To the best of our knowledge, the *computation reuse* mechanism is a novel contribution that has not been explored in NAT, and has demonstrated great promise and effectiveness.
>
> **Secondly**, while our *"disentangled architecture"* may appear similar to MAE & MAGE, they are in fact different.
> Specifically, MAE & MAGE view the idea of encoding only visible tokens as a way to accelerate the training process for *representation learning*.
> In contrast, our *disentangled design*  focuses on building an efficient architecture tailored for *image generation*.
> The distinct objective results in small but important methodological differences, primarily manifested in the decoder's attention.
> Specifically, both MAE and MAGE adopt full attention on all tokens ([M] & [V] → [M] & [V]) in their decoder.
> On the contrary, the decoder in ENAT only inputs the mask token and integrates the visible ones via SC-Attention, creating a tailored "[M] & [V] → [M]" information flow in the decoder.
> Experimental results show that naively adopting MAE/MAGE-type full attention only increases computational overhead without enhancing performance:
>
> | **Decoder Attention Type** |**Decoder Attention Information Flow** | **GFLOPs↓** | **FID↓** |
> |-|----------------------|---------------------|----------|
> | MAE & MAGE type |[M] & [V] → [M] & [V]             | 24.5                | 5.03     |
> | Ours               | [M] & [V] → [M]|**22.6**                | **4.97**     |
>
> This aligns with our observation in Section 4.1: In NAT generation, visible tokens primarily provide information for mask tokens. Thus, the [V] → [M] part in [M] & [V] → [M] has adequately fulfilled the role of the visible token, making it unnecessary to incorporate the visible token into the decoder input and perform full attention.
>
> We will add more discussions to clarify the relationships with these works in our revision.
>
> >**W2: Intuition for Enabling Smaller Sampling Steps**
>
> The key intuition behind our better performance with smaller steps is that: each decoding step in ENAT is more effective. To illustrate this, we conducted an experiment by applying different mask ratios (mimicking inputs in different steps of generation) to ground truth images and measuring the quality of *one-step decoded results* (i.e., the model directly predicts all masked positions in one forward pass) using FID.
>
> |  **Mask Ratio** | 0.2 | 0.4 |0.6 | 0.8 |
> |--------------|-------|-------|--------|--------|
> | MaskGIT (Baseline)       |  3.89 | 8.23 | 19.20 | 48.78 |
> | ENAT (Ours)     |  **3.48** | **6.74** | **16.04** | **45.51** |
> | $\Delta$     |  0.41 | 1.49 | 3.16   | 3.27   |
>
> As the results show, ENAT consistently outperforms the baseline NAT design. This suggests that ENAT's predictions for masked positions are more reliable (i.e., of higher quality) at every decoding step, allowing them to take larger strides (decode more tokens per step) and producing high-quality generation with fewer refinement steps.
>
> The qualitative results for the 0.2 mask ratio setting and the intermediate generation results of ENAT are also visualized in **Fig. 2** and **Fig. 1** of our global response to further illustrate this point.

---

### Official Review · Reviewer_PoGn · 2024-07-16

**Soundness:** 4
**Presentation:** 3
**Contribution:** 4
**Rating:** 6
**Confidence:** 4

**Summary:**

This paper proposes an improved Efficient MaskGIT-like non-autoregressive image generation method. Its core component is to (1) spatially disentange visible tokens and mask tokens in attention, and prioritize compute for the visible tokens; (2) temporally reuse the visible tokens from previous decoding step. The results on common benchmarks like ImageNet and MSCOCO demonstrate the effectiveness of the proposed approach, leading to clear FID improvement while significantly reducing computational cost.

**Strengths:**

The idea is straightforward and makes intuitive sense to me. Since the mask tokens do not really convey any meaningful information, they do not have to be in the attention operation with visible tokens. Also, as the visible tokens decoded from the prior steps remain unchanged, their features should be reusable in the next decoding step. This paper is generally well-written and relatively easy to follow. The results, especially on ImageNet, show promising improvement.

**Weaknesses:**

I cannot help but notice the comarison between U-ViT and ENAT on MSCOCO (Table 5). An 8-step U-ViT performas almost as good as ENAT-B, although on ImageNet ENAT is clearly the better choice. In Table 5, the number of parameters of U-ViT is smaller but its FLOPs is larger compared with ENAT-B, so I guess one could say they are in the same level. Does that indicate that ENAT is not as advantageous on text-to-image synthesis task?

**Questions:**

1. In line 221-229, the authors mention the modifications of training pipeline to accomodate the need for reusing previous features. I wonder if the authors could elaborate on that? My understanding is that this MaskGIT paradigm is a single-step process during training and multi-step process during decoding? So where does the features of prior steps come from during training? And how is that related to the masking modification mentioned in line 222?
2. If my understanding is correct, the spatial disentanglement is also a compute-efficient operation, as there is no more attention across visible tokens and mask tokens in most of the layers. Yet in most experiments, the authors describe it as a performance-boosting design (say, Table 6(a)). I wonder is it because the authors have stacked more attention layers to compensate for the FLOPs loss? If so, the authors are encouraged to further clarify their setup.

---

> ### Author Rebuttal · Authors · 2024-08-05
>
> > **W1: Questions on MSCOCO Results**
>
> **Firstly**, we kindly clarify that the two models, U-ViT and ENAT-B, may *not* be viewed as being "on the same level."
> Our primary objective is to improve inference efficiency.
> Therefore, FLOPs, as an important indicator of practical efficiency (see Fig. 6 of our original paper), is much more critical than model parameter count.
> Moreover, the increase in model parameters is in fact an intentional result of our disentangled architecture, which significantly improves performance *without* affecting efficiency (see Tab. 6a of our original paper).
>
> **Secondly**, we have only trained our ENAT model on COCO text-to-image generation for 150K steps (see A.2 of our original paper) as it already outperforms U-ViT trained with nearly 7$\times$ more training budget (i.e. 1000K steps).
> In a fairer comparison where U-ViT is also trained for 150K steps, ENAT is still significantly better:
> | Model  | Train Steps | Generation Steps | TFLOPs↓ | FID↓  |
> |--------|-------------|------------------|--------|------|
> | U-ViT  | 1000K        | 8                | 0.5    | 6.92|
> | U-ViT  | 150K        | 8                | 0.5    | 10.02|
> | ENAT-B | 150K        | 8                | **0.3**    | **6.82** |
>
> We will add more discussions on this in our revised experiment section.
>
> > **Q1: Elaboration on ENAT's Training Process**
>
>   Sure, the detailed process of training in reuse mode (Eq. (5)) is elaborated below:
>   1. **Masking Tokens**: Given the current input token map $\mathbf{v}$, we mask a random subset (*e.g.*, 30%) of visible tokens in $\mathbf{v}$ to create $\mathbf{v}^{\text{prev}}$.
>   2. **Feature Extraction**: Feed $\mathbf{v}^{\text{prev}}$ into the NAT model to obtain its features $\mathbf{z}^{\text{prev}}$.
>   3. **Forward Pass and Loss Computation**: Use Eq. (5) to forward the current input token map $\mathbf{v}$ along with $\mathbf{z}^{\text{prev}}$ obtained in Step 2, and compute the loss. In practice, a stop gradient operation is applied before feeding $\mathbf{z}^{\text{prev}}$ into Eq. (5).
>
> Since $\mathbf{v}^{\text{prev}}$ created in Step 1 contains less visible tokens than $\mathbf{v}$, it effectively simulates the "previous step input" that ENAT will encounter during generation, thereby equipping ENAT with the ability to correctly handle previous step's feature for reuse.
>
> We will elaborate this process in more detail in our revision.
>
> > **Q2: Setups for Spatial Disentanglement Experiments**
>
> Yes, we intentionally kept the FLOPs unchanged for a clear comparison. However, we achieved this by enlarging the network width, which we will further clarify in our revised manuscript. Enhanced performance at the same FLOPs generally equates to lower computational cost for the same performance, thus still demonstrating our method's efficiency.

---

> > ### Comment · Reviewer_PoGn · 2024-08-12
> >
> > Thanks for the response! I have no more questions.

---

### Author Rebuttal · Authors · 2024-08-07

We thank all the reviewers for their insightful comments and suggestions.

We are encouraged that our work was found to be **well-motivated/written** (all reviewers), with our findings **offering fresh new perspectives** (Reviewer bqgC) and our method being **simple** (Reviewer GVb9) and **effective** (Reviewer PoGn, GVb9, BoDK).

One common concern was the need for more visualized results to elucidate why ENAT works well or to demonstrate its effectiveness more comprehensively. To address this, we have included qualitative analyses (**Fig. 1,2**) and many more samples (**Fig. 4,5,6**) of our method in the attached PDF file. To avoid confusion, the figures in the rebuttal pdf are all referred to in **bold**.
More qualitative results will also be incorporated in our revision.

Other specific concerns are elaborated in the responses to each reviewer. We hope our responses appropriately address your concerns.

---

### Author Response · Authors · 2024-08-12

Dear Reviewers,

Thank you once again for your valuable comments and questions, which have greatly contributed to improving our work! As the rebuttal period draws to a close (ending on August 13), we would be very grateful if
you could share any additional questions or concerns whenever it is convenient for you.

We truly appreciate your further discussions and are actively available here to address any further questions
you may have.

Looking forward to hearing back from you!

---

### Decision · Program_Chairs · 2024-09-25

**Decision:**

Accept (poster)

**Comment:**

The submitted manuscript proposes to improve MaskGIT-like non-autoregressive image generation models by spatially disentangling
 visible tokens and mask tokens during self-attention in order to prioritize visible tokens and to reuse tokens from previous decoding steps - both modifications towards increased efficiency. The modifications have beneficial effect on FID and model efficiency.
All reviews agree that the paper is well written - also, the idea is overall simple yet effective.
Some initial criticism regarding method details and ablations (for example the difference to MAE and MAGE) were clarified during the rebuttal phase so that the paper has overall positive scores after the  discussion phase.